SOFTWARE

# Playbook workflow builder: Interactive construction of bioinformatics workflows

**Daniel J.B. Clarke[1], John Erol Evangelista[1], Zhuorui Xie[1], Giacomo B. Marino[1], Anna I. Byrd[1], Mano R. Maurya[2], Sumana Srinivasan[2], Keyang Yu[3], Varduhi Petrosyan[3], Matthew E. Roth[3], Miroslav Milinkov[4], Charles Hadley King**  **[5], Jeet Kiran Vora[5], Jonathon Keeney[5], Christopher Nemarich[6,7], William Khan[6,7], Alexander Lachmann[1], Nasheath Ahmed[1], Alexandra Agris[1], Juncheng Pan[1], Srinivasan Ramachandran[2], Eoin Fahy[2], Emmanuel Esquivel[3], Aleksandar Mihajlovic[4], Bosko Jevtic[4], Vuk Milinovic[4], Sean Kim[5], Patrick McNeely[5], Tianyi Wang[5], Eric Wenger[6,7], Miguel A. Brown[6,7], Alexander Sickler[6,7], Yuankun Zhu[6,7], Sherry L. Jenkins[1], Philip D. Blood[8], Deanne M. Taylor[6,7], Adam C. Resnick[6,7], Raja Mazumder[5], Aleksandar Milosavljevic[3], Shankar Subramaniam[2], Avi Ma'ayan[1]***

**1** Department of Pharmacological Sciences, Windreich Department of Artificial Intelligence and Human Health, Mount Sinai Center for Bioinformatics, Icahn School of Medicine at Mount Sinai, New York, New York, United States of America, **2** Department of Bioengineering, University of California San Diego, La Jolla, California, United States of America, **3** Department of Molecular and Human Genetics, Baylor College of Medicine, Houston, Texas, United States of America, **4** Persida Inc., Brooklyn, New York, United States of America, **5** Department of Biochemistry and Molecular Medicine, The George Washington School of Medicine and Health Sciences, Washington, DC, United States of America, **6** Department of Biomedical and Health Informatics; Department of Pediatrics, The Children's Hospital of Philadelphia, University of Pennsylvania Perelman School of Medicine, Philadelphia, Pennsylvania, United States of America, **7** Center for Data Driven Discovery in Biomedicine, The Children's Hospital of Philadelphia, Philadelphia, Pennsylvania, United States of America, **8** Pittsburgh Supercomputing Center, Carnegie Mellon University, Pittsburgh, Pennsylvania, United States of America

* avi.maayan@mssm.edu

## Abstract

The Playbook Workflow Builder (PWB) is a web-based platform to dynamically construct and execute bioinformatics workflows by utilizing a growing network of input datasets, semantically annotated API endpoints, and data visualization tools contributed by an ecosystem of collaborators. Via a user-friendly user interface, workflows can be constructed from contributed building-blocks without technical expertise. The output of each step of the workflow is added into reports containing textual descriptions, figures, tables, and references. To construct workflows, users can click on cards that represent each step in a workflow, or construct workflows via a chat interface that is assisted by a large language model (LLM). Completed workflows are compatible with Common Workflow Language (CWL) and can be published as research publications, slideshows, and posters. To demonstrate how the PWB generates meaningful hypotheses that draw knowledge from across multiple resources, we present several use cases. For example, one of these use cases prioritizes drug targets for individual cancer patients using data from the NIH Common Fund programs GTEx, LINCS, Metabolomics, GlyGen, and ExRNA. The workflows created with PWB can be repurposed to tackle similar use cases using different inputs. The PWB platform is available from: https://playbook-workflow-builder.cloud/.

**Data availability statement:** There are no primary data in the paper; the project's source code with instructions is available from: https://doi.org/10.5281/zenodo.14641815 and https://github.com/MaayanLab/Playbook-Workflow-Builder. The Playbook Workflow Builder platform is available from: https://playbook-workflow-builder.cloud/.

**Funding:** This project was funded by NIH grants: OT2OD036435 to A.Ma. and S.S. (CFDE Workbench), OT2OD030160 to A.Ma. (LINCS DCC), OT2OD030544 to S.S. (MW DCC), OT2OD030547 to A.Mi. (ERCC DCC), OT2OD030162 to A.C.R. (KF DCC), and OT2OD032092 to R.M. (GlyGen DCC). The funders had no role in study design, decision to publish, or preparation of the manuscript.

**Competing interests:** The authors have declared that no competing interests exist.

## Introduction

The rate of growth in diversity and volume of biological and biomedical data is rapidly increasing. This rapid growth poses challenges for our ability to discover, access, interoperate, integrate, and analyze these data. Although hundreds of bioinformatics tools and databases are published each year, few efforts aim to organize and integrate all these resources into integrative platforms. As biological and biomedical data analyses become increasingly more complex and customized, and at the same time more standardized, workflow engines and workflow languages that combine tools and databases are needed [1–5]. Although there are many bioinformatics workflow engines and workflow languages (S1 Table), each of these resources has advantages and disadvantages. Broadly, workflow engines modularize data analysis tasks into steps that can be performed in isolation. Capturing dependencies between each step facilitates stringing them into workflows. Most workflow engines and workflow languages are task-agnostic and operate at the command-line interface (CLI).

Some of the first generation workflow platforms geared towards bioinformatics were Ruffus [6], Anduril [7,8], Bioconductor workflows [9], and Taverna [10,11]. Ruffus and Anduril are Python libraries that make it easier to combine analysis from multiple tools. Taverna was a larger project that was initially called Taverna Workbench and later Apache Taverna. It could be operated as a desktop application, CLI, or via a remote execution server. Taverna was coupled with a catalog of workflows called BioCatalogue [12]. With the arrival of the cloud and due to rapid expansion in the availability of bioinformatics tools, the original platforms such as Ruffus and Taverna were superseded with platforms that offered more features and flexibility. These platforms are led by Galaxy [13–17] an internationally large-scale well-funded project that offers many features including a user interface (UI), a library of components, and extensive user training. Alternatives to Galaxy include platforms such as Snakemake [18,19] and NextFlow [20].

Two leading community standardized workflow languages are Common Workflow Language (CWL) [21] and Workflow Description Language (WDL) [22], both of these standards decouple the workflow specification from the task management and execution. CWL can be executed by cloud workspaces that implement the Global Alliance for Genomics and Health (GA4GH) Workflow Execution Service (WES) API [23]. Two leading cloud platforms that implement WES are CAVATICA [24] and Terra [25]. Other examples of community standards developed to encode metadata about workflows include BioCompute Objects, a JavaScript Object Notation (JSON) schema validatable IEEE standard (IEEE 2791-2020) that is published in the BioCompute Object portal repository [26], and WorkflowHub [27]. WorkflowHub describes workflows by adopting the Research Object Crate (RO-Crate) standard [28] leveraging schema entities from BioSchemas [29].

Besides the workflow engines mentioned above, there are also many efforts for facilitating the construction of bioinformatics analyses by combining tools to perform various parts of a scientific report. For example, BioConductor [9] is a repository of curated R packages for bioinformatics tools that can interoperate with one another. The thousands of publicly available bioinformatics tools and databases with APIs gave rise to another class of systems tangentially related to workflow engines. These are federated knowledge graphs (KGs). Examples of such systems include the BioThings Explorer [30] which invokes APIs documented and registered with the SmartAPI registry [31] to dynamically resolve edges between two destination data types. BioThings Explorer is related to the Translator project [32,33] which operates similarly but with a UI. caGrid [34], BioMoby [35], and Mobyle [36] are platforms for constructing API-based workflows and the Semantic Automated Discovery and Integration (SADI) [37] is a way to annotate bioinformatics web-services with semantics for the purpose of API interoperation.

Another class of workflow engines are bioinformatics applications that enable users to upload their data into a cloud environment and then select from a collection of tools to produce a single workflow that produces a report. For example, BioJupies [38] is a workflow engine for performing RNA-seq analysis in the cloud. Users of BioJupies can start with a gene count matrix, or raw FASTQ files. After these files are uploaded, the user can pick from a collection of tools that will be executed in a Jupyter Notebook to produce a report that resembles a publication [39]. The BioJupies platform was later extended to enable analysis of many other data types with Appyters [40]. Appyters are parameterized Jupyter Notebooks converted into full-stack web-based applications. Other similar platforms include GenePattern [41] and iLINCS [42]. Several of the platforms in this category interoperate with the workflow languages CWL and WDL.

Since its inception in 2004, the US National Institutes of Health (NIH) Common Fund (CF) has funded more than 50 large-scale biomedical research programs. CF programs have generated large and diverse datasets with the aim of having these datasets propel biomedical research forward by serving as resources for hypothesis generation and integrative systems level analyses. These datasets include various omics profiling from across thousands of human subjects, cell lines, organoids, and animal models. Each CF program typically has a Data Coordination Center (DCC) that is tasked with managing these datasets and serving them to the community via databases, tools, workflows, and search engines. CF DCCs typically develop data portals that serve the raw data from their respective CF program, as well as providing more processed knowledge extracted from such data. To accomplish this, CF DCCs developed tools that enable users to interactively explore datasets via user interfaces as well as via well-documented APIs.

However, enabling knowledge discovery by combining data and tools from multiple CF programs remains both a challenge and an opportunity. To address this challenge, the NIH established the Common Fund Data Ecosystem (CFDE) consortium (https://cfde.info). In its first phase, the CFDE consortium established a data model that standardizes cross-program data elements such as genes, tissues, drugs, and diseases [43]. These harmonized identifiers can be used to find data files produced by multiple CF programs, but such data model fails to directly enable cross-program hypothesis generation. Here we demonstrate how by leveraging data, tools, and well-documented APIs from multiple CF programs, and other sources, we constructed a visual user-friendly web-based workflow construction platform called the Playbook Workflow Builder (PWB). In contrast with other bioinformatics workflow engine platforms, PWB requires stricter annotations and specifications of workflow building-block components that were curated to facilitate bioinformatics data knowledge discovery leveraging CF primary datasets and tools. Such extensive descriptions of components, termed meta-nodes, enable complex data analyses that result in complete reports that resemble research publications. Moreover, users of the PWB can interactively and visually construct workflows by exploring all possible available options at each workflow building step. Alternatively, users can interact with a chatbot interface to query the PWB workflow building blocks with a text prompt to automatically produce new workflows.

## Design and implementation

### The knowledge resolution graph (KRG)

To dynamically develop workflows that draw knowledge from across bioinformatics tools and databases, we organized well-documented APIs into an integrative network of microservices. Nodes in this network represent semantic types, for example, variants, genes, glycans, metabolites, drugs, gene sets, gene expression signatures, and diseases. Edges in the network

represent operations performed by various tools applied to these semantic types. For example, enrichment analysis applied to a set of genes, principal component analysis (PCA) applied to a data matrix, or a PubMed search query with a search term that describes a disease. Nodes and edges in the network are characterized in a strict type-safe manner forming a programmatically defined data structure. We term this network a knowledge resolution graph (KRG). In contrast to a KG, a KRG encodes the capacity of obtaining knowledge by means of some computational or manual process. In other words, instead of subjects connected via predicates in a KG, a KRG has functions connected via common data types. Knowledge obtained from one tool or database may be augmented, compared, or supplemented with knowledge from another. The KRG can be used to find compatible processes with instantiated knowledge at any step when constructing a workflow. The edges in the KRG are mostly API-driven microservices providing interoperability across bioinformatics tools and databases. The APIs must be documented with OpenAPI [44] and deposited into the SmartAPI [31] repository. Such compliance with these standards eases implementation.

The assembled metanodes are then used to facilitate a collection of use cases and use case templates. Use case templates are defined as workflows with the same structural components but with application to different data instances. For example, gathering information about a gene or a variant from several databases can be done for a single gene, but also as a template that supports the querying of other genes by changing the input query. The collected use cases are geared toward accumulation of evidence from transcriptomics, metabolomics, glycomics, proteomics, epigenomics, genomics, and other assay types. The workflows that are generated for realizing these use cases are reusable and extendible. To enable access to the system, a user-friendly interface (UI) was developed. The UI is geared towards experimental biologists with no programming background. The PWB system is set up in a way that other developers can contribute to the system, and/or reuse components of the PWB for enhancing their own web portals and bioinformatics data analysis workflows. Metanodes are accessible via a uniform REST API that supports multi-step workflow executions via CWL. Thus, the KRG graph can be queried programmatically.

## Metanode specifications

Metanodes are workflow building block components that are specified with TypeScript. The specification captures common identifiable metadata elements about each workflow building block component. The metadata about a component includes human-readable labels, descriptions, icons, authors, license, and versioning information. The specification then couples these semantics with type-safe implementations which inherit types from dependent components. A metanode can be of three types: a *prompt*, a *resolver*, or a *view* (Fig 1). A *view* function renders the interactive visualization of an instance of the type of interface. A *resolver* function accepts one or many *data types* as inputs and produces a single *data type* as an output. A *prompt* is an interactive React component that can accept input *data types* to facilitate decisions made by the user for transforming the inputs into a single output *data type*, for example, selecting a gene from a list, or submitting a gene set for enrichment analysis. With these three types of metanodes, we can construct workflows. A *prompt* with no inputs can inject an initial instance of a *data type* object, and that instance can be used as an input argument to compatible *resolvers* or *prompts* to yield other *data type* instances, or figures, tables, and charts. Metanodes also specify parts of a story. This is a parameterized sentence about what that component is doing. This sentence will appear in the methods section of the output report. This sentence is written in a style that is typical for a Methods section of a research paper including citations. These sentences are chained together into paragraphs to construct a human-readable description of the entire workflow. The paragraphs can be further reorganized and copyedited using an LLM like GPT-4 [45].

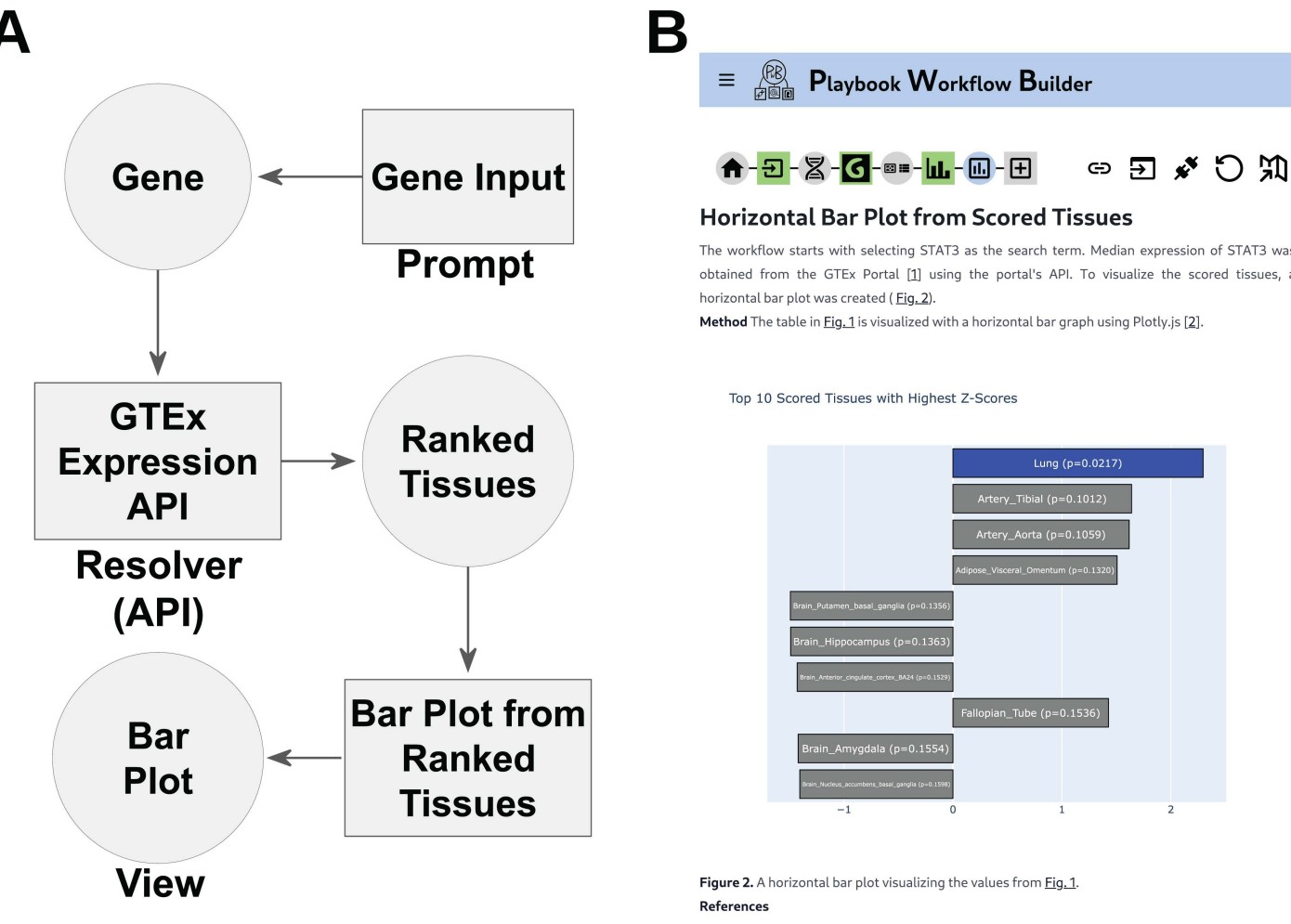

**Fig 1. The different PWB metanode types are strung together to form workflows.** A. In this example, the *prompt* type of metanode takes a gene as the input; then the *resolver* metanode uses the GTEx API to obtain the expression of the input gene from across human tissues. Finally, a *view* metanode visualizes the contents returned from the API as a bar chart. B. Screenshot from the executed workflow in the PWB platform.

## System modularity

Because of the metanode specification, PWB metanodes can be developed, tested, and operated independently from the PWB codebase. All the implemented metanodes are collected and assembled into a unified KRG database (Fig 2). The PWB system queries and utilizes this database to construct the data-driven UI (Fig 3). As such, the PWB web-based application is a product of the contents of the KRG database, and thus, extending the functionality of the PWB web-based application only requires creating and registering additional metanodes. By modularizing the PWB processes, we can mix, match, and stack PWB metanodes to construct parameterizable workflows. PWB metanodes and workflows have consistent interfaces and can thus be exposed in consistent ways such as over API, in CWL workflows, or through additional interfaces.

## Fully persistent process resolution graph (FPPRG) database

While the KRG can be used to construct arbitrary workflow templates, a workflow is an instance of that template operating on a unique dataset. To store data from a workflow, an

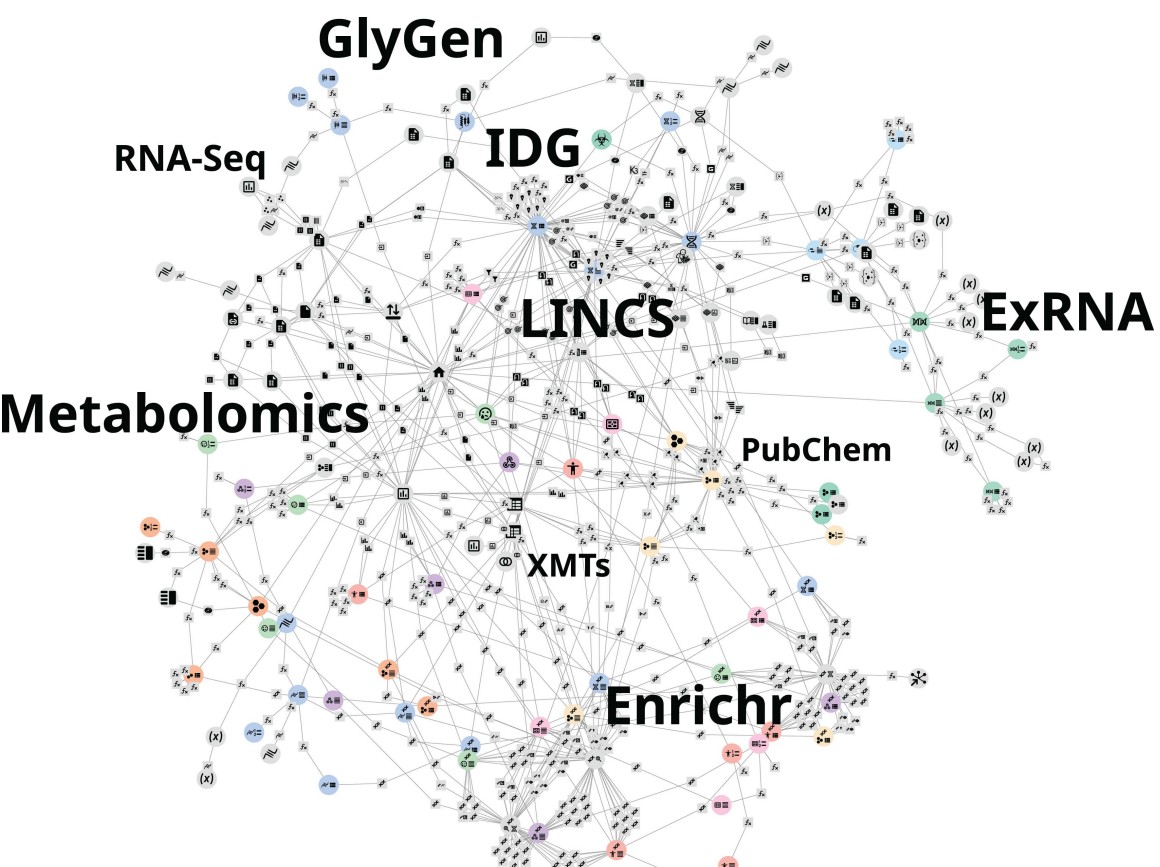

**Fig 2. Network visualization of the PWB knowledge resolution graph (KRG).** The network of connected metanodes is interactive and can be explored from the user interface.

additional database is established. This additional database stores the data that flows through workflows. As such, the database ensures collision-free updates and a self-deduplication. Another feature of this additional database is the decoupling of workflow templates from the actual data that flows through those workflows, providing further deduplication and reproducibility. In this database, executed workflows are stored in 4 tables (Fig 4). The first table is a dependency graph of each constructed step of a workflow. This information is stored in a record called a *Process*. This record is tightly coupled with the *Component*, it stores the *Component* ID, a JSON object for *Prompt* configuration, and back references to any other *Process* whose output is used by this record. The second table is a fully persistent list (FPL) [46]. It stores sequential order of a workflow through a linked list. A singular list can be resolved with the ID of the last element of the list, and each intermediary state has a unique ID. Importantly, elements of the lists need only be stored once even if used in multiple lists. The third table is a *Result* record. It has a one-to-one relationship with a *Process* record and is constructed by performing the execution using the function from the *Component* type referenced in the *Prompt*. Finally, the 4th table is a *Data* record. This table contains JSON Binary Large Objects (BLOBs) used to store data in the *Process* and *Result* tables. All IDs are created by hashing the content of the record. A unique series of user steps can be stored and accessed by a single ID through the FPL, while the dependency graph ensures deduplication of the workflows regardless of order. Finally, the actual results of any given workflow step are stored. Requests for the output

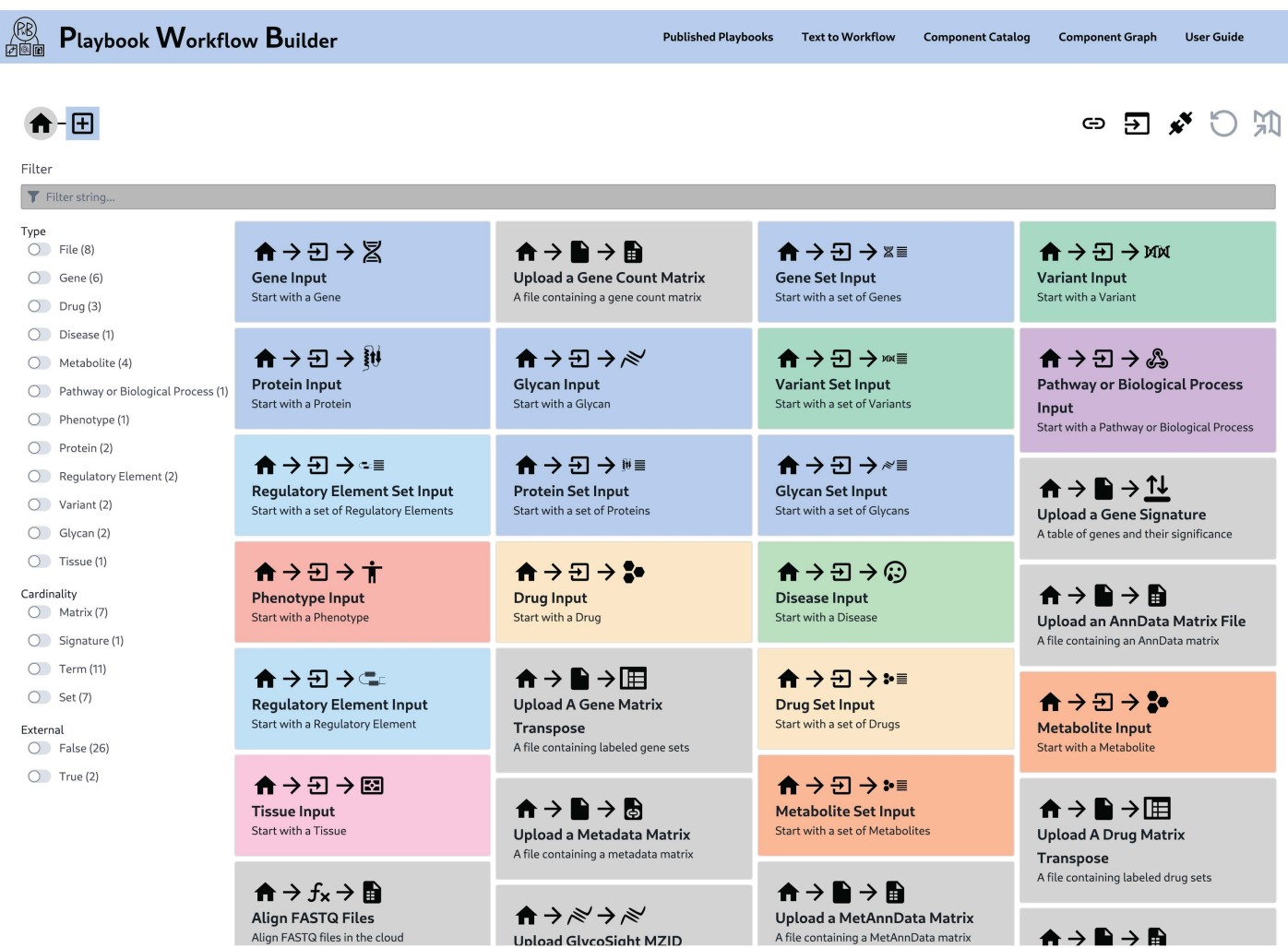

**Fig 3. The landing page of the PWB UI provides access to a collection of *prompt* metanodes to begin constructing workflows.**

of any *Process* are sent to a queue of workers if the *Result* does not already exist. Hence, steps are executed simultaneously if there are enough workers, and equivalent execution results are deduplicated. Altering an earlier step in a workflow can be done with a git-style rebase. A new FPL and dependency graph starting from the parent of the modified node are created and expanded to the previous tail. *Result* records would then be computed as required to obtain the new output for the entire workflow.

## Developing the PWB user interface

The PWB UI is developed in TypeScript with Next.js, a full-stack framework that uses React and offers isomorphic server-side and client-side rendering. Tailwind CSS-based DaisyUI and Blueprint.js are used for styling the site and data tables. NextAuth.js is used for managing user accounts via ORCID or e-mail. The FPPRG database, which stores workflow executions can operate entirely in memory or with a PostgreSQL database in a production setting. Workers run in the main process or execute independently on different machines. Message passing is achieved through PostgreSQL's listen/notify feature. The website's navigation and metanode rendering are driven by queries to the in-memory KRG over REST API or WebSocket. The

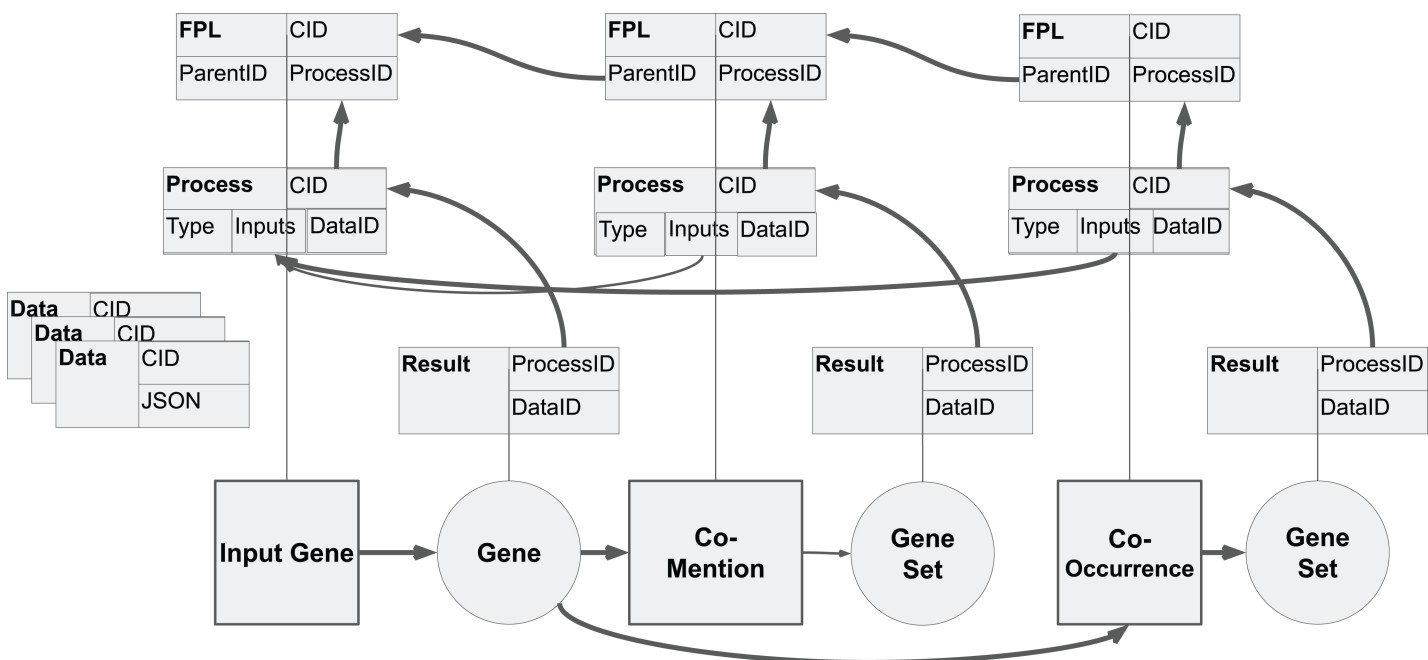

**Fig 4. The structure of the Persistent Process Resolution Graph (FPPRG) Database.** The FPPRG database stores the data that flows through workflows in four tables. The first table is a dependency graph of each constructed step of a workflow. The second table stores the sequential order of a workflow. The third table is a *Result* record, and the 4th table is a *Data* record.

UI is decoupled from the metanodes facilitating the independent development of the website and the metanodes. This also means that a completely new set of metanodes can be used for a platform with a different focus. All metanodes' TypeScript, Python, and other dependencies are assembled and installed into a single Docker container. This container is used to run the PWB workers. A smaller Docker container with only JavaScript dependencies runs the UI.

## Cloud agnostic file storage

A Python library was developed to help with managing files in a storage system that is agnostic to the cloud provider. All files uploaded to the PWB are stored and accessed using an abstract layer provided by this library. In development, files are stored on the local disk, while in production, the files are stored in an S3 bucket. Alternatively, users can have their files in a CAVATICA workspace [24] when CAVATICA sessions are established. Once uploaded, files are stored by their SHA-256 checksum which provides content-based addressing for deduplication. An entry is added to the database and is associated with the user who uploaded the file. These records receive universally unique identifiers (UUID) and are served by the PWB platform using GA4GH's Data Repository Service (DRS) protocol [47]. Files on the platform are then treated as DRS URIs which can be resolved anywhere in the system. Files can also be provided to the platform directly from external DRS hosting platforms. Functional helpers are available to obtain the contents of the DRS files/bundles or for uploading new files from PWB metanodes.

## Workflow format translations

The FPPRG database encodes workflows along with their data. The steps of the workflow are encoded in the KRG where metadata about each step can be resolved. These steps can be translated and exported to other workflow description formats providing interoperability with

other platforms. Hence, the PWB platform provides users with the ability to export constructed workflows into several workflow specification standards. These are outlined below:

**BioCompute Objects.** Establishing better conceptual descriptions of workflows is important for reproducibility [48]. Workflow languages are machine-readable files that should be able to provide all the details needed to re-execute workflows. However, usually there is insufficient information that is needed to fully reproduce a workflow. For this, the IEEE 2791-2020 BioCompute Objects (BCO) standard was developed [26]. BCO is a rigorously defined standard for bioinformatics analysis workflow documentation that is flexible enough to accommodate any pipeline, but rigid enough to define a structure for computable metadata to annotate workflows. There is an ecosystem of tools that are compatible with the BCO standard, including cloud genomics platforms like Seven Bridges Genomics, CAVATICA, and DNAnexus. The BioCompute Portal is part of this ecosystem, and acts as a repository of published BCOs, as well as a place to manually build BCOs. The portal is provided with several published examples [49,50]. Using the PWB interface, a BCO can be constructed from any workflow. The exported BCO contains full provenance about the workflow including description of the steps, data, versioning, and authorship. The serialized BCO specifications can be downloaded as well as submitted to the BioCompute Portal via API where they can be inspected, modified with additional annotations, or extended to other schemas, and ultimately published.

**Common workflow language (CWL).** Common Workflow Language (CWL) is an open standard for describing how to run command line tools and connect them to create workflows [21]. A command line interface (CLI) was developed from the KRG to invoke any *Process* metanode, providing inputs in JSON-serialized files, and writing the output to a JSON-serialized file. Using this CLI, a CWL *CommandLineTool* specification can be constructed out of any *Process* metanode, and a CWL *Workflow* specification and input variables file can be constructed from the FPPRG database. License, authorship, versioning information, and descriptions for the individual metanodes in use, are included in the resulting CWL output. All *Prompt* data that may be captured via interactions between the user and the UI are instead specified in the input variable file. Hence, the PWB platform metanodes are fully compatible with CWL, and CWL workflows can be exported from the PWB interface.

**Research object crate (RO-Crate).** RO-Crate is a community-based specification for research data packaging of Research Objects (RO) with rich metadata, based on open standards and vocabularies including JSON Linked Data (JSON-LD) and schema.org [51]. Adopting a similar structure to describing workflows as WorkflowHub [52], an RO-Crate can be created from entries in the FPPRG database containing complete workflow provenance including license, authorship, versioning information and descriptions for the individual metanodes in use. The RO-Crate can then be used for registering PWB workflows in WorkflowHub and for minting citable Digital Object Identifiers (DOIs) for published workflows.

**External API-driven metanodes.** The PWB platform is agnostic to the actual implementation of each step but a significant portion of the currently implemented metanodes rely on external API services. These services can more efficiently provide the most up-to-date knowledge as the data behind these APIs evolves. The downside of this is that reproducibility of workflows in not guaranteed. To mitigate this risk, we annotate the relevant steps that rely on external services and save the outputs of those services with a timestamp when the execution occurred.

## Constructing workflows from prompts with an LLM

The user interface of the PWB facilitates construction of workflows by presenting to the user all possible next steps compatible with the current step. This functionality is also presented as a prompt to a large language model (LLM) chatbot. A generative pre-trained transformer

(GPT) model is tasked with making decisions about the best next step to take when presented with a text prompt from the user. Using a few-shot prompt, we direct the chatbot to choose from a set of possible next steps based on user messages. We accept single suggestions automatically and present multiple suggestions to the user. Selected suggestions are included in an incrementally constructed workflow and rendered in a chat box-style interface along with LLM Assistant messages. Because we use the chatbot assistant to only help build a PWB workflow based on the constrained KRG, the risk of hallucination is mitigated. In the worst case, users may receive a self-documented workflow that performs some analysis that is not intended. By collecting feedback from users in the form of thumbs up and down, we plan to fine-tune the model to provide increasingly accurate workflows based on user prompts.

## Results

The Playbook Workflow Builder (PWB) is a web-based interactive workflow construction platform. The workflow engine facilitates user traversal through a network of microservices stored in a knowledge resolution graph (KRG). The metanodes include well-documented functions or API wrappers that are executed on-demand with the inputs of the previous step to produce the outputs for the next; and React components that render interactive visualizations using the output of the previous step and producing the outputs for the next step based on user interaction. The PWB user-friendly web-based interface facilitates users to extend, branch, and merge workflows that are executed while they are constructed. Users can construct workflows manually by clicking on cards and breadcrumbs, or via a chatbot interface. Notably, the PWB system provides the means to modify workflows on-the-fly while all past versions of the workflows remain persistent [46]. PWB workflows are saved and accessible via fixed URLs. This makes any user session a reusable and reproducible workflow template. Each step in the workflow can be inspected and extended, and the entire workflow can be viewed as a complete Jupyter Notebook-style report.

Besides constructing new workflows, PWB users can start with published workflows created by others. The published workflows contain detailed descriptions of each step, and this provides the ability to construct reports that resemble research publications. These published workflows can be re-executed by uploading new user data. Once the user uploads their own data, and when users adjust a workflow, a new workflow is created and executed, and the results are presented and available via a unique URL. The automatically generated text with citations describing the workflow is created by stacking descriptions from each step. This description can be reformatted and simplified with an LLM or adjusted manually by the user to enhance and customize it. Some features of the platform require users to log in. These features include uploading files, saving workflows, publishing workflows in the PWB catalog of published workflows, contributing suggestions, using the LLM features, and publishing workflows as BioCompute Objects or operating the playbook within CAVATICA.

### Implemented metanodes

The PWB platform provides users with the ability to perform a wide variety of analyses powered by the network of metanodes. These metanodes are used as steps in workflows. So far, we have developed 561 such metanodes (S2 Table). Below we describe some of the currently implemented metanodes.

**RNA-seq data analysis and visualization.** Beginning from a user-uploaded count matrix of gene expression, where each row represents a gene, and each column is a sample with associated metadata, data is uploaded to the PWB and encoded with AnnData [53]. From the gene expression matrix, several metanodes enable different normalization and

data visualizations. For example, the contents of the data matrix can be visualized with PCA [54], UMAP [55], or t-SNE [56] plots. These metanodes are supported by the Scanpy Python package [57]. From the data matrix, users can compute differential expression to produce gene expression signatures. Differential expression analysis can be performed with methods such as the Characteristic Direction [58], limma-voom [59,60], or DESeq2 [61]. Differentially expressed genes can be used as input for downstream analysis such as enrichment analysis which is described next.

**Enrichment analysis.** Enrichment analysis can be performed within the PWB using the Enrichr API [62]. The gene sets *data type* in the PWB can be submitted for enrichment analysis against the many gene set libraries available from Enrichr. For example, the GTEx [63] and ARCHS4 [64] tissue expression gene set libraries can be selected to obtain ranks of more relevant tissues. Similarly, the KEGG [65] and WikiPathways [66] gene set libraries can be used to prioritize relevant cell signaling pathways. Enrichr also provides an API to search for metadata terms across the Enrichr gene set libraries. For example, a disease term search can be used to construct a consensus gene set [67]. Another way to obtain gene sets is through literature search. By submitting to PubMed any search term, term-gene co-mentions in publications can be used to construct the most relevant gene set for any search term. This functionality is supported by PWB via the Geneshot API [68].

**Gene set manipulation.** The gene matrix transpose (GMT) file format is commonly used to serialize gene set libraries. GMT files contain lists of terms followed by sets of genes associated with each term. GMT files can be uploaded and analyzed by the PWB platform. A common way to interrogate the overlap between several gene sets is through UpSet plots [69] or SuperVenn diagrams [70]. The PWB has metanodes to display interactive versions of such plots. Additionally, several operations are implemented to transform *data types* from one to another. For example, turning ranked lists of genes into gene sets by choosing a cutoff, turning multiple gene sets into a GMT file, or collapsing a GMT file into a single gene set by applying a consensus or a union set operation on it.

**Healthy human tissue expression atlases.** GTEx has profiled postmortem tissues from healthy donors with RNA-seq to produce gene expression data matrices from 54 human tissues [63]. The GTEx API can be used to find median tissue expression levels for all human genes for each one of these 54 profiled tissues. Similarly, the ARCHS4 resource [64] was created by uniformly aligning approximately 2 million publicly available RNA-seq samples collected from human and mouse. The ARCHS4 API can also be used to find median tissue expression across over 200 tissues and cell types. The PWB enables users to obtain summary statistics from these APIs, which can be visualized as bar graphs. It is also possible to use these data resources as a baseline to identify novel drug targets. For example, gene expression data collected by RNA-seq from tumor samples, can be compared to all normal tissue to identify genes that are only highly expressed in the tumor using the TargetRanger API [71].

**Metanodes created from LINCS resources.** The Library of Integrated Network-Based Cellular Signatures (LINCS) program [72] profiled the response of human cells to thousands of chemical and genetic perturbations followed by omics profiling. The PWB provides several metanodes related to prioritizing drugs and preclinical small molecules for targeting individual genes and gene expression signatures. For example, a metanode can be used to perform LINCS L1000 reverse search queries for a given gene, producing interactive visualizations and tables of significant LINCS L1000 chemical perturbagen signatures that may maximally increase or decrease the expression level of the single human gene. A similar metanode was implemented to provide search against the L1000 CRISPR KO signatures. Other metanodes enable users to query the SigCom LINCS database [73] with gene expression signatures or gene sets. Such signatures may be in the form of a vector of differential gene

expression, or up- and down-regulated gene sets. Both types of input signature queries can yield ranked lists of chemical perturbations and CRISPR KOs.

**Metanodes created from GlyGen resources.** GlyGen is an international initiative funded by the NIH to promote research about glycoscience [74]. The GlyGen consortium developed a web-based portal that brings together glycan and protein specific data from major resources such as UniProt [75], GlyConnect [76], Protein Data Bank (PDB) [77], UnicarbKB [78], ChEBI [79] and PubChem [80] and other resources [81]. These datasets are presented to users through a standardized data model [82] via the GlyGen data portal (https://data.glygen.org). The GlyGen API endpoints (https://api.glygen.org) facilitate the same functionality provided by the user interface, providing the PWB with several GlyGen metanodes that can be integrated in various workflows. The GlyGen metanodes also support data visualization and kinase enrichment analysis. Furthermore, the GlyGen metanodes operate several core data types such as, glycans, proteins, and glycoproteins. For other glycoconjugate species, such as glycolipids, GlyGen metanodes implement the passthrough search APIs to the GlySpace alliance [83] and other resources. In addition, uploaded mass spectrographic glycan files are analyzed with various GlyGen specific metanodes, and then knowledge is extended with other PWB metanodes.

**Metanodes created from metabolomics resources.** The Metabolomics Workbench (MW) is another resource supported by the NIH CF [84]. MW contributed several metanodes to the PWB including those from the bioinformatics tools MetGENE [85], MetENP [86], and a gene ID conversion tool. These tools, originally designed to be stand-alone web applications, provide REST APIs to obtain relevant information for analyses related to profiled metabolites within the PWB. MetGENE is a hierarchical, knowledge-driven tool designed for gene-centered information retrieval. By entering a single gene, or a set of genes, users can access information related to the gene such as pathways, reactions, metabolites, and studies from metabolomics in MW. To refine searches, MetGENE incorporates filtering options based on organism, tissue or anatomy, and disease or phenotype. This feature provides tailored and context-specific search experience. Several metanodes using MetGENE are implemented that take as input either a gene, or a gene set, for downstream analyses. The relevant functionality from MetENP is provided via a REST API called MetNet. Briefly, given a list of metabolites, e.g., metabolites with significant change between two conditions such as disease/normal or treatment/control in a metabolomics study obtained by using MetENP or another tool, a researcher may want to find what are the pathways and functions affected. MetENP/MetNet facilitates metabolite name harmonization using RefMet [87], metabolite class enrichment, metabolic pathway enrichment and visualization, and identification of reactions related to the given metabolites and genes coding for enzymes catalyzing these reactions. In MetNet, the list of these genes can be used to develop their protein-protein interaction (PPI) subnetwork using the STRING database APIs [88]. Each of these metanodes has an associated table that renders the information obtained from the API.

**The connect the dots (CTD) metanode.** The Connect the Dots (CTD) metanode takes as input a set of genes or proteins and identifies a subset of genes or proteins that are highly connected within either knowledge graphs or networks derived from gene expression, metabolomic or other omic datasets [89]. CTD algorithm has previously discovered multi-gene biomarkers of drug response to breast cancer therapies based on mouse PDX models [90], and metabolomic signatures of rare inborn errors of metabolism [89,91]. While CTD has been previously deployed as independent R and Python packages (https://github.com/BRL-BCM/CTD), its deployment on the Playbook will allow for its use by a wider scientific audience. The CTD workflow starts with an input set of genes. The user then has the option of identifying significant connections within this set in the STRING protein-protein

interaction network [88], WikiPathways [66], or a network derived from user-supplied data. The networks represented as weighted graphs, can be derived from expression data, proteomic data, metabolomics, or any other normalized omic dataset. This allows for users to identify highly connected sets of genes within their specific disease, treatment, or condition of interest. Given a weighted graph and a set of graph nodes as an input, CTD identifies significant highly connected subsets. An optional "guilt by association" feature identifies neighboring nodes using probability diffusion. CTD also returns a visual display of the nodes and connections.

**Metanodes created from ERCC resources.** The ExRNA Communication Consortium (ERCC) Common Fund (CF) Data Coordination Center created a framework and toolset for FAIR data, information, and knowledge that delineate the regulatory relationships between genes, regulatory elements, and variants, and made them available to PWB via metanodes. We have implemented the ClinGen Allele Registry (CAR) and Genomic Location (GL) Registry [92], variant and genomic region on demand naming services, respectively. The CAR canonical identifiers (CAid) or Genomic Location identifiers (GLid) provided are reference genome-agnostic, stable, and globally unique. The ERCC metanodes enable the retrieval and mapping of unique identifiers and other commonly used identifiers, such as dbSNP IDs [96], connected through the Allele Registry and GL Registry using the Allele Registry RESTful APIs. Moreover, we have created the CFDE Linked Data Hub (LDH) [93], a graph-based database, to extract and link tissue and cell type-specific regulatory information from SCREEN [94], GTEx [63], and other CF projects, including Roadmap Epigenome [95] and EN-TEx [96]. Each excerpt on the CFDE LDH is created in a machine-readable format and contains a link to the original data source for provenance tracking. The CFDE LDH RESTful APIs provide read and write capabilities for both accessing and contributing gene regulatory information. This enables the CFDE LDH to connect more than 800 million regulatory data and information documents, which can be quickly retrieved by PWB through the API endpoints given any variant, regulatory region, or gene as input.

## The book of use cases

The PWB currently contains a collection of fully implemented and published workflows. These workflows were first designed by drawing each workflow as a flowchart diagram (S1 Fig). Each flowchart represents a unique workflow contributed by different groups that worked collaboratively on the project. In these diagrams, each node represents a metanode. Each flowchart representing a workflow also lists the name of the workflow and the resources used to obtain the data needed to run the workflow. The color of each metanode was used to track the status of the implementation of each metanode and the entire workflow. The flowchart plots were used as a guide to capture ideas about potential workflows. Thus, not all these designed workflows are fully implemented. In some cases, implemented workflows do not match exactly the flowchart diagram that was used to design it.

## Use case workflow templates and workflow instances

The PWB fully implemented and published workflows are listed on a dedicated area on the PWB site termed the PWB catalog of workflows (Fig 5 and Table 1). Each published workflow has a title, a short description, a description of the inputs and outputs, the data resources used, the authors, version, license, the date of publication, and a button to launch the workflow. Since each workflow is parameterized, we consider these workflows as templates. These templates can be executed with different inputs to produce a new workflow. Below we describe several selected published PWB workflows in detail.

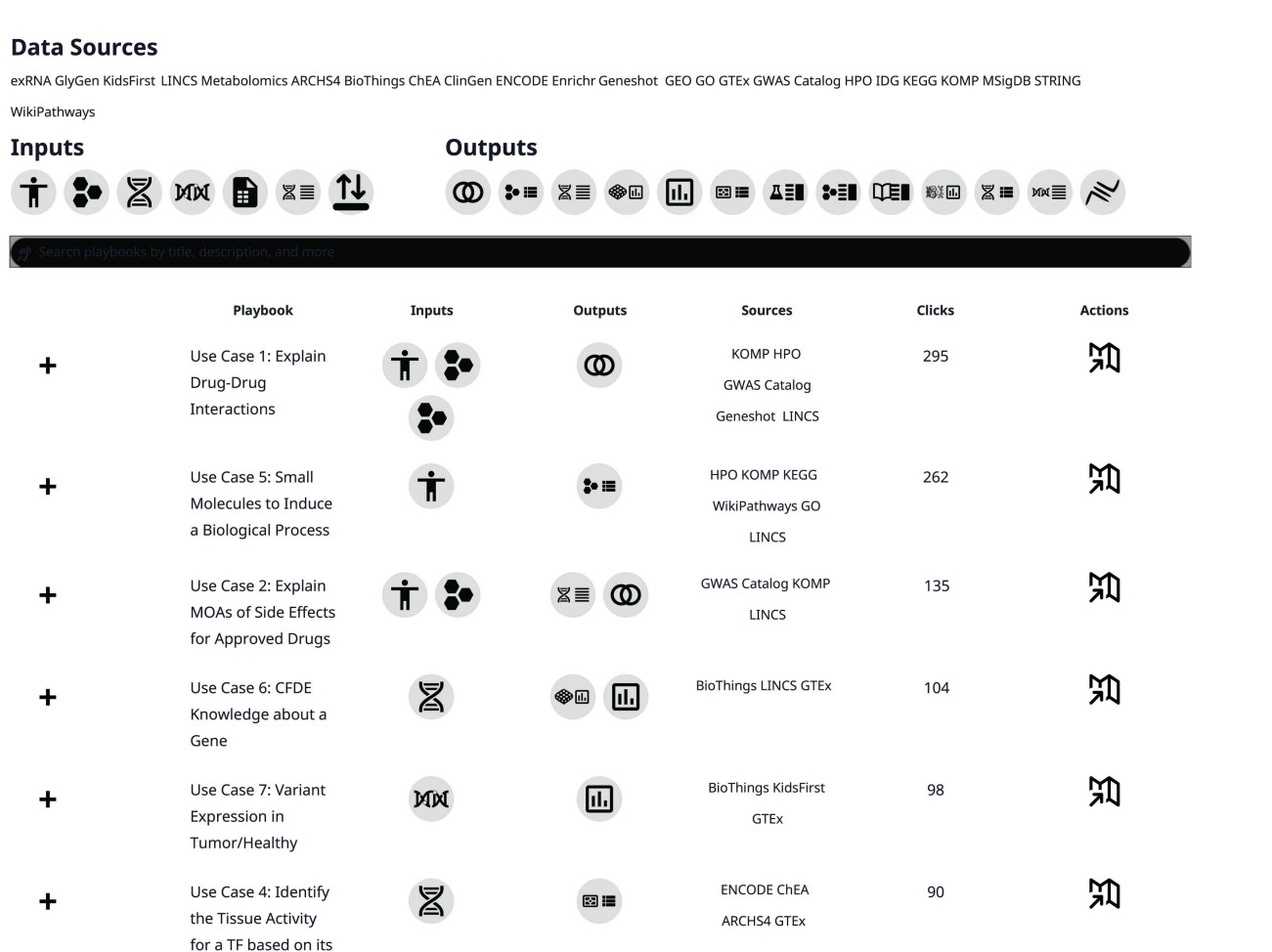

**Fig 5. Published workflows are curated workflows that are listed on a dedicated page that catalogs these in a table.** Each workflow entry can be expanded to obtain more information about the workflow and to launch the workflow within the PWB platform in report mode.

**Use case 13: Cell surface targets for individual cancer patients analyzed with Common Fund datasets.** The input to this workflow is a data matrix of gene expression that was collected from a pediatric tumor from the Kids First CF program [24]. The RNA-seq samples are the columns of the matrix, and the rows are the raw expression gene counts for all human coding genes. This data matrix is fed into TargetRanger [71] to screen for targets that are highly expressed in the tumor but lowly expressed across most healthy human tissues based on gene expression data collected from postmortem patients with RNA-seq by the GTEx CF program [63]. Based on this analysis, the gene Insulin-like growth factor II m-RNA-binding protein 3 (IMP3) was selected because it was the top candidate returned from the TargetRanger analysis (Table 2). Next, we leveraged unique knowledge from various other CF programs to examine knowledge related to IMP3. First, we queried the LINCS L1000 data [97] from the LINCS program [72] converted into RNA-seq-like LINCS L1000 Signatures [98] using the SigCom LINCS API [73] to identify mimickers or reversers small molecules and CRISPR KOs that maximally impact the expression of IMP3 in human cell lines. These

**Table 1.  List of published use cases available for re-execution and expansion on the PWB platform. \*The base URL for the DOIs is "10.48546/WORKFLOWHUB.WORKFLOW.".**

| Label | DOI* | Inputs | Output |
|---|---|---|---|
| Use Case 1: Explain Drug-Drug Interactions | 1237.3 | Phenotype; Drug; Drug | SuperVenn Visualization |
| Use Case 2: Explain MOAs of Side Effects for Approved Drugs | 1238.2 | Phenotype; Drug | Gene Set; Supervenn Visualization |
| Use Case 3: Compounds to Reverse Disease Signatures | 1248.2 | Gene Signature; Gene Signature | Scored Drugs |
| Use Case 4: Identify the Tissue Activity for a TF based on its Targets | 1239.2 | Gene | Scored Tissues |
| Use Case 5: Small Molecules to Induce a Biological Process | 1240.2 | Phenotype | Scored Drugs |
| Use Case 6: CFDE Knowledge about a Variant | 1241.2 | Variant | LINCS L1000 Reverse Search Dashboard; Plotly Plot |
| Use Case 6: CFDE Knowledge about a Gene | 1242.2 | Gene | LINCS L1000 Reverse Search Dashboard; Plotly Plot |
| Use Case 7: Variant Expression in Tumor/Healthy | 1243.2 | Variant | Plotly Plot |
| Use Case 9: Identifying regulatory relationships between genes, regulatory regions, and variants | 1249.3 | Variant | Regulatory Element Set |
| Use Case 10: Guilt by Association | 1244.2 | Gene Set | Gene Set |
| Use Case 11: Related Proteins/ Metabolites across DCCs | 1245.2 | Gene | MetGENE Reaction Table; MetGENE metabolite table; MetGENE Studies Table |
| Use Case 13: Novel Cell Surface Targets for Individual Cancer Patients Analyzed with Common Fund Datasets | 1246.2 | Gene Count Matrix | LINCS L1000 Reverse Search Dashboard; MetGENE Summary; Scored Genes; Scored Drugs; MetGENE Reaction Table; MetGENE metabolite table; Variant Set; GlyGen Protein Products |

**Table 2.  Ranked list of targets identified by TargetRanger to be highly expressed in the tumor sample and lowly expression across normal tissues from GTEx.**

| Gene | Z-score |
|---|---|
| IMP3 | inf |
| ARHGDIA | inf |
| GPRIN1 | 7.23 |
| CARM1 | 6.98 |
| JSRP1 | 6.70 |
| SLC7A6 | 6.60 |
| NBPF15 | 5.78 |
| RABGEF1 | 5.76 |
| HPS4 | 5.64 |
| ANKRD39 | 5.21 |

potential drugs and targets were filtered using the CF IDG program's list of understudied proteins [99] to produce a set of additional targets. Next, IMP3 was searched for knowledge provided by the Metabolomics Workbench MetGENE tool [85]. MetGENE aggregates knowledge about pathways, reactions, metabolites, and studies from the Metabolomics Workbench CF supported resource [84]. The Metabolomics Workbench was searched to

find associated metabolites linked to IMP3. Furthermore, we leveraged the Linked Data Hub (LDH) API [93] to list knowledge about regulatory elements associated with IMP3. Finally, the GlyGen database [74] was queried to identify relevant sets of proteins that are the product of the IMP3 genes, as well as known post-translational modifications discovered on IMP3. The discovery of IMP3 is not completely novel, IMP3 has been previously reported to be aberrantly expressed in several cancer types and its high expression is associated with poor prognosis [100].

**Use case 1: Explaining drug-drug interactions.**  This workflow takes as input an adverse event term and two drugs. The adverse event is identified in several databases that contain gene sets already associated with the adverse events and mammalian phenotypes related to the adverse event. Namely, matching adverse events and mammalian phenotypes are identified from the GWAS Catalog [101], MGI Mammalian Phenotype Ontology [102], and from the Human Phenotype Ontology (HPO) [103]. A set of consensus genes associated with the matching terms is assembled. Then, the workflow queries the LINCS L1000 chemical perturbation signatures [73] with the two input drugs to find gene sets that are consistently up- or down-regulated by the treatment of human cell lines with these drugs. The consensus gene sets impacted by the drugs, and the gene set related to the adverse events are then compared and visualized using a SuperVenn diagram to highlight overlapping genes between these sets. Genes of interest are those affected by both drugs and are associated with the phenotype. Such overlapping genes can be further interrogated individually for evidence in the literature, or as a gene set using enrichment and network analyses.

To demonstrate the workflow for a specific instance, we start with the adverse event "bleeding" and the drugs warfarin and aspirin. It is known that these drugs interact to increase the risk of internal bleeding [104] but the exact intracellular mechanism of such interaction is still not fully understood. The workflow starts with selecting "bleeding" as the search term. Gene sets with set labels containing the word "bleeding" were queried from Enrichr [1]. Identified matching terms from the GWAS Catalog 2019 [2], MGI Mammalian Phenotype Level 4 2019 [3] and the Human Phenotype Ontology [4] libraries are then assembled into a collection of gene sets. A GMT file is extracted from the Enrichr results for all the identified gene sets from each library and then these are combined using the union set operation. Gene sets with set labels containing the terms warfarin and aspirin were next identified from the LINCS L1000 Chem Pert Consensus Sigs [5] library. The gene sets collected for each drug were combined into one gene set library. The collection of gene sets was then visualized with a SuperVenn diagram (Fig 6). This analysis identified 243 genes up-regulated and 245 genes down-regulated by warfarin; 249 genes up-regulated and 244 genes down-regulated by aspirin, 85 genes associated with bleeding from MGI, and 35 from HPO. Only one gene, namely THBS2, is up regulated by both drugs, and is also associated with bleeding related phenotype in MGI. While the gene SLC7A11 is downregulated by both drugs and is linked to an MGI bleeding phenotype. THBS2 is a member of the thrombospondin family, and as such it plays a critical role in coagulation. It was shown that knockout mice of THBS2 have an increased bleeding time phenotype (MP:0005606) [105] and THBS2 is a potent inhibitor of tumor growth and angiogenesis [106]. It is difficult to explain why both drugs are found to up-regulate this gene. The expected effect is that these drugs would reduce the expression of the genes to reduce coagulation. At the same time, both drugs are also found to down-regulate the expression of the amino acid transporter SLC7A11. SLC7A11 knockout mice also have an increased bleeding time phenotype (MP:0005606), and mutations in this gene have been implicated in many acute human diseases through induction of ferroptosis [107,108]. Hence, for SLC7A11 the direction of the impact of the drugs on its expression is consistent with other prior evidence.

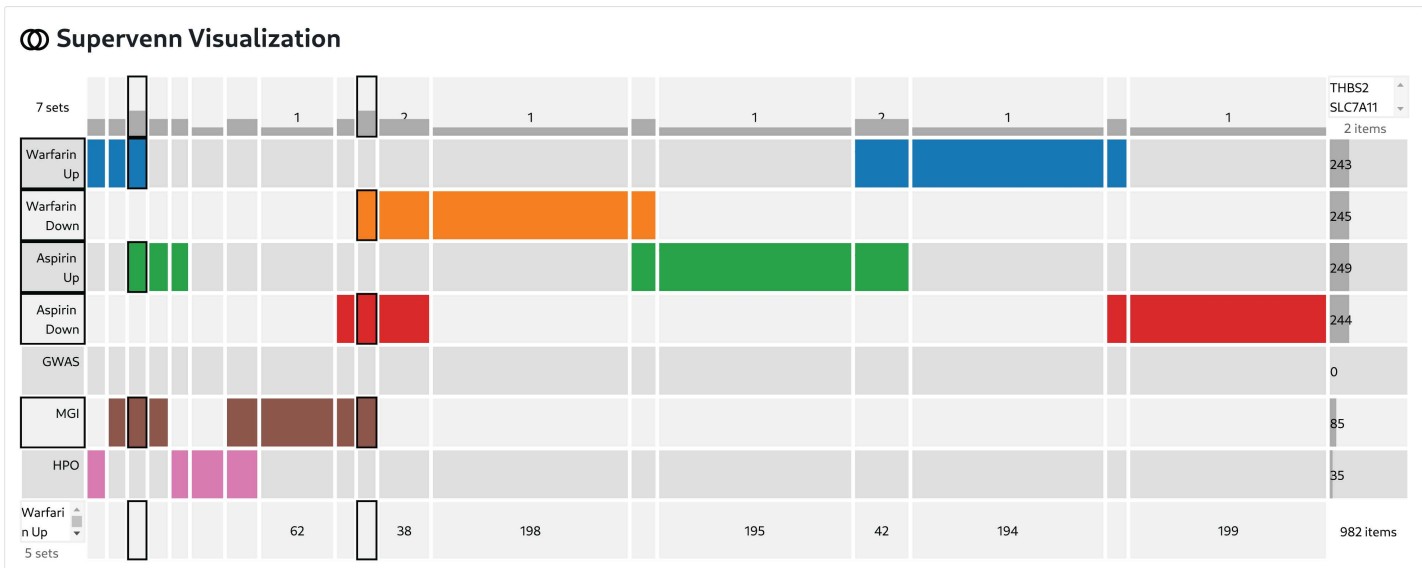

**Fig 6. SuperVenn diagram to visualize the overlap between sets of genes that are up and down regulated by aspirin and warfarin based on LINCS L1000 signatures, as well as knockout mouse, HPO, and GWAS phenotypes associated with the term "bleeding".** The permanent URL for a description of this workflow is: https://doi.org/10.48546/WORKFLOWHUB.WORKFLOW.1237.3.

**Use case 11: Related proteins/metabolites across DCCs.** The enzyme ribulose-5-phosphate epimerase (RPE) participates in the catalysis of the interconversion of ribulose-5-phosphate (Ru5P) to xylulose-5-phosphate (Xu5P) in the pentose phosphate pathway. A recent study [109] focused on the biophysical and enzymatic characterization of RPE in several organisms. Interestingly, the study suggested that RPE may play a crucial role in protection against oxidative stress. Toward integrative analysis to further elucidate the roles of RPE in various pathways and mechanisms of human disease, we collected knowledge about PRE from various NIH CF programs and other sources. The collected information about RPE includes: 1) Associated metabolites from the Metabolomics Workbench [84]; 2) Expression across human tissues from GTEx [63]; 3) Small molecules and single gene knockouts that maximally induce the expression of RPE from LINCS [73]; 4) Associated variants from ClinGen via LDH [110]; 5) Protein-protein interactions from STRING [88]; and 6) Regulation of RPE by transcription factors from ChEA3 [111]. In addition, the use case converts RPE into a gene set using the Geneshot API [68]. The Geneshot API returns a set of 100 genes that mostly correlate with RPE based on thousands of human RNA-seq uniformly processed from GEO [112]. Co-expression correlations computed from the data processed by ARCHS4 [64]. The comprehensive approach to find knowledge about a single gene is also applied to the generated gene set with all six resources. The final report provides a mechanistic understanding of how RPE can affect various pathways and functions despite not being involved in the pathways and processes directly.

**Use case 10: Identifying gene regulatory relationships between genes, regulatory elements, and variants.** This workflow takes as input one or more genes, regulatory elements, or variants. One may then query for regulatory relations of the selected entity type with other entity types. In one application, we may ask what genomic regions regulate a gene of interest and what evidence supports that regulatory relationship. We start the workflow by providing the gene of interest as input. We first focus on regulatory elements that are in the vicinity of the gene body identified using the epigenomic data from NIH Roadmap

Epigenomics [95] and ENCODE projects stored in the ENCODE SCREEN database [94]. Regulatory evidence associated with the SCREEN regulatory elements was connected to genes and variants using CFDE LDH [93], a graph-based database that facilitates the linking of findable, accessible, interoperable, and reusable (FAIR) [113] information about genes, regulatory elements, and variants retrieved through well-documented RESTful APIs. The available regulatory information includes: 1) Variants associated with regulatory elements from the ClinGen Allele Registry [92]; 2) Allele-specific epigenomic signatures, such as DNA methylation, histone modifications, and transcription factor binding, from Roadmap Epigenomics [95] and EN-TEx [96] projects; 3) Quantitative trait loci information from GTEx [63] and other studies; and 4) Regulatory element activity, all presented in a tissue- and cell-type-specific manner. The workflow also provides users with commonly used identifiers for variants that fall within a regulatory element of interest, including those from dbSNP [114], ClinVar [115], and the ClinGen Allele Registry [92].

## System scalability

To test the ability of the platform to horizontally scale to support simultaneous users, we simulated many parallel users submitting workflows via the API with increasing depth and with an increasing number of horizontally scaled backend processes. Workflow depth refers to the maximum number of workflow steps where each step depends on the results of a prior step. Workflows with more depth are inherently slower to resolve and are not parallelizable. Overall, the horizontal scaling of the PWB platform should be able to serve at least 50 concurrent users with our currently allocated resources (S2 Fig).

## Reproducibility of workflows

The PWB platform was designed to empower biomedical researchers without coding skills. Users can find and invoke bioinformatics workflows by leveraging the availability of public datasets, and commonly used data visualization methods. In contrast with other workflow engines, PWB users are provided with written details with citations about steps of the analysis. The PWB platform was initially created for a specific purpose of bringing to the surface datasets and tools developed in projects that received support from the NIH Common Fund. Due to this initial goal, many of the PWB components trigger external APIs and this feature of the platform has advantages and disadvantages. The use of federated microservices enables tool and repository owners to continuously improve and update these services independently of the PWB platform. Most repositories and web-based bioinformatics tools typically have the computational infrastructure to resolve queries in real-time, assembling all the data behind the service. Hence, it may become cost prohibitive for a single researcher, or a single platform to locally host these resources in one place. Additionally, many resources developed by Common Fund supported programs continue to evolve as additional data is gathered. The maintenance of these APIs and the data these APIs query pose a long-term sustainability challenge. While the PWB platform cannot guarantee the reproducibility of the results because federated knowledge continues to evolve, the PWB platform supports exporting reports. These exports contain the workflow, the output produced by every step in the workflow, and a timestamp of when the APIs were called. This description of the workflow can be interrogated, and the entire workflow can be imported into a Docker-served container using the same version of the original APIs. It is important to note that PWB components are not inherently required to be external microservices. In fact, roughly half of the PWB components are not. These components operate without external APIs. Most PWB components that access large and evolving indexed databases, however, are wrappers around external APIs. The nature of evolving

federated services makes assembling all parts of all workflows into an isolated environment is largely intractable. As the field rapidly evolves in data types and methods, out-of-date tools inevitably become less useful. Importantly, the PWB platform retains timestamped results so that reports persist, but re-execution of these reports is subject to change, providing the most recent knowledge from each microservice.

## Comparison to other platforms

In contrast with Galaxy [13], and the Galaxy ToolShed [116], PWB metanodes operate with semantic and runtime asserted JSON intermediaries instead of files. These artifacts make API usage more practical, and it enables custom rendering of information passed throughout the system. PWB components can also be implemented to receive user input, and components can have interactive visualizations. These features provide customization via user interaction. Examples of customized user interactions that are not possible with Galaxy, but present in PWB, include: autocomplete for user-input selection, extracting regions from an interactive set overlap visualization, and interactive labeling of columns from a data table. Additionally, constructing a Galaxy workflow involves selecting from a massive list of all possible Galaxy supported tools and configuring each tool by providing it with all needed options and files. In contrast, PWB workflows start with a selection of a data file, or other information provided by the users. Once such selection is made, only the applicable tools for the next step become available. This happens at each step of workflow construction. Each user step selection in the PWB environment yields a new workflow capable of executing, exporting, saving, and sharing with a persistent URL.

Considering other relevant platforms, Bioconductor [9] annotates packages using biocViews, a direct acyclic graph (DAG) made of terms extracted from a controlled vocabulary which helps users locate packages relevant for a particular analysis. caGrid [34], BioMoby [35], and Mobyle [36] had syntactic and semantic interoperability but these platform are no longer available. PWB has annotated functions with annotated input arguments and return types, these annotations form a Knowledge Resolution Graph (KRG) which can be used to enumerate all compatible functions with a given type. Semantic types are used for disambiguating standard software types, for example, a gene symbol or a drug name rather than a string, or a gene count matrix, rather than just a file. The Semantic Automated Discovery and Integration (SADI) [37] is a way to annotate bioinformatics web-services functions with semantics. This makes SADI compatible with the PWB's KRG functions. In practice, however, most services do not provide SADI, and typically require minor tweaks to be compatible with other services. The PWB platform's centralized approach allows APIs to be annotated without requiring modifications to the APIs. Furthermore, PWB components can also facilitate the definition of user-facing form-controls and interactive visualizations, something that none of the other platforms currently facilitate.

Overall, the PWB platform is independent of the workflow *execution* system. This means that an existing workflow tool, like snakemake or CWL runner, could be employed with no change to the user experience. Currently, unresolved executions are queued in a postgreSQL database, and a horizontally scalable set of workers execute the tasks as they come in, while skipping tasks with unresolved dependencies. The PWB platform's primary focus is guiding the user through the analysis steps where provenance is fully captured. This produces a reusable workflow. For computationally demanding jobs, and for working with patient protected private data, users of the PWB should execute their workflows via the CAVATICA integration. In CAVATICA, users can run any computationally intensive workflow with any GA4GH compatible workflow execution service. The CAVATICA integration can be used to reduce the burden of utilization of PWB native pooled resources available for the community of most PWB users.

## Availability and future directions

There are no primary data in the paper; the project's source code with instructions is available from: https://doi.org/10.5281/zenodo.14641815 and https://github.com/MaayanLab/Playbook-Workflow-Builder.

The Playbook Workflow Builder platform is available from:
https://playbook-workflow-builder.cloud/

In the future, we plan to continue to expand the capabilities of the PWB platform based on user feedback and community adoption. So far, we have conducted three use-a-thon events that introduced the PWB to new users. These users provided valuable feedback about the platform that we fully implemented. The users in these three events were both "regular" users that worked with the PWB via the UI, and "power" users that contributed new PWB components. The "regular" users provided feedback on how to improve the UI, while the "power" users provided feedback on the documentation and SDK for contributing PWB components.

So far, most of the metanodes and use cases implemented by the PWB platform are related to systems biology, molecular networks, and the analyses of genes, variants, metabolites, and post-translational modifications. The platform is extensible and could be applied to other areas of biomedical research domains such as structural biology, cheminformatics, genomics, and clinical research. In addition, the PWB platform can be applied in other domains besides biomedical research. The chat interface of the PWB also opens opportunities for applications that may enhance the functionality of chatbots in general. By executing workflows on demand to produce knowledge and deeper understanding, chatbots that currently are mostly based on large language models (LLMs) could be enhanced with a PWB-like system operating in the background to construct knowledge from building blocks.

## Supporting information

**S1 Fig. Workflow diagrams created to design various workflows.** Before implementing metanodes to construct workflows, the workflows were sketched as flowchart diagrams. Note that not all nodes and links in the diagrams were implemented exactly how they were designed.
(PDF)

**S2 Fig. Horizontal scaling of the platform workflow executions.** Users submitting workflows in parallel were simulated with the platform using between 1 and 50 parallel users submitting workflows of depth 5 and 10. Figure shows the time in seconds it took for the entire workflow to be completed for each individual simulated user (A), and the time it took to submit that workflow via the API (B). Simulations performed with 1, 2, and 4 horizontally scaled backend processes each with 5 worker threads.
(PDF)

**S1 Table. Bioinformatics Workflow Platforms.** A list of only a few key related bioinformatics workflow engines with various features compared across platforms.
(XLSX)

**S2 Table. Implemented PWB Metanodes.** Listing of 561 currently implemented metanodes with detailed descriptions of their types, inputs and outputs, name, description, and text to add to the report about each step.
(XLSX)

## Acknowledgments

The authors would like to thank the NIH CFDE team and the useathons and workshops participants for their support and suggestions.

## Author contributions

**Conceptualization:** Daniel J.B. Clarke, Avi Ma'ayan.

**Data curation:** Anna I. Byrd.

**Formal analysis:** Daniel J.B. Clarke, John Erol Evangelista.

**Funding acquisition:** Deanne M. Taylor, Adam C. Resnick, Raja Mazumder, Aleksandar Milosavljevic, Shankar Subramaniam, Avi Ma'ayan.

**Methodology:** Daniel J.B. Clarke, Avi Ma'ayan.

**Project administration:** Matthew E. Roth, Jeet Kiran Vora, Christopher Nemarich, Aleksandar Mihajlovic, Eric Wenger, Sherry L. Jenkins, Avi Ma'ayan.

**Software:** Daniel J.B. Clarke, John Erol Evangelista, Zhuorui Xie, Giacomo B. Marino, Mano R. Maurya, Sumana Srinivasan, Keyang Yu, Varduhi Petrosyan, Miroslav Milinkov, Charles Hadley King, Jeet Kiran Vora, Jonathon Keeney, William Khan, Alexander Lachmann, Nasheath Ahmed, Alexandra Agris, Juncheng Pan, Srinivasan Ramachandran, Eoin Fahy, Emmanuel Esquivel, Bosko Jevtic, Vuk Milinovic, Sean Kim, Patrick McNeely, Tianyi Wang, Miguel A. Brown, Alexander Sickler, Yuankun Zhu, Avi Ma'ayan.

**Supervision:** Matthew E. Roth, Jeet Kiran Vora, Aleksandar Mihajlovic, Sherry L. Jenkins, Philip D. Blood, Deanne M. Taylor, Adam C. Resnick, Raja Mazumder, Aleksandar Milosavljevic, Shankar Subramaniam, Avi Ma'ayan.

**Visualization:** Daniel J.B. Clarke, John Erol Evangelista, Zhuorui Xie, Giacomo B. Marino, Mano R. Maurya, Sumana Srinivasan, Keyang Yu, Varduhi Petrosyan, Jeet Kiran Vora.

**Writing – original draft:** Daniel J.B. Clarke, Mano R. Maurya, Sumana Srinivasan, Keyang Yu, Varduhi Petrosyan, Charles Hadley King, Jonathon Keeney, Avi Ma'ayan.

**Writing – review & editing:** Avi Ma'ayan.

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
