## [Decision Letter · Decision Letter 0]

21 Nov 2024

PCOMPBIOL-D-24-01433Playbook Workflow Builder: Interactive Construction of Bioinformatics WorkflowsPLOS Computational Biology Dear Dr. Ma'ayan, Thank you for submitting your manuscript to PLOS Computational Biology. After careful consideration, we feel that it has merit but does not fully meet PLOS Computational Biology's publication criteria as it currently stands. Therefore, we invite you to submit a revised version of the manuscript that addresses the points raised during the review process. Please submit your revised manuscript within 60 days Jan 20 2025 11:59PM. If you will need more time than this to complete your revisions, please reply to this message or contact the journal office at ploscompbiol@plos.org. Please include the following items when submitting your revised manuscript: * A rebuttal letter that responds to each point raised by the editor and reviewer(s). You should upload this letter as a separate file labeled 'Response to Reviewers'. This file does not need to include responses to formatting updates and technical items listed in the 'Journal Requirements' section below. * A marked-up copy of your manuscript that highlights changes made to the original version. You should upload this as a separate file labeled 'Revised Manuscript with Track Changes'. * An unmarked version of your revised paper without tracked changes. You should upload this as a separate file labeled 'Manuscript'. If you would like to make changes to your financial disclosure, competing interests statement, or data availability statement, please make these updates within the submission form at the time of resubmission. Guidelines for resubmitting your figure files are available below the reviewer comments at the end of this letter. We look forward to receiving your revised manuscript. Kind regards, Pedro MendesSection EditorPLOS Computational Biology Feilim Mac GabhannEditor-in-ChiefPLOS Computational Biology Jason PapinEditor-in-ChiefPLOS Computational Biology **Journal Requirements:** 1) Your manuscript is missing the following sections: Design and Implementation, and Availability and Future Directions. Please ensure that your article adheres to the standard Software article layout and order of Abstract, Introduction, Design and Implementation, Results, and Availability and Future Directions. For details on what each section should contain, see our Software article guidelines:https://journals.plos.org/ploscompbiol/s/submission-guidelines#loc-software-submissions 2) Please upload all main figures as separate Figure files in .tif or .eps format. For more information about how to convert and format your figure files please see our guidelines: https://journals.plos.org/ploscompbiol/s/figures 3) Some material included in your submission may be copyrighted. According to PLOSu2019s copyright policy, authors who use figures or other material (e.g., graphics, clipart, maps) from another author or copyright holder must demonstrate or obtain permission to publish this material under the Creative Commons Attribution 4.0 International (CC BY 4.0) License used by PLOS journals. Please closely review the details of PLOSu2019s copyright requirements here: PLOS Licenses and Copyright. If you need to request permissions from a copyright holder, you may use PLOS's Copyright Content Permission form.Please respond directly to this email and provide any known details concerning your material's license terms and permissions required for reuse, even if you have not yet obtained copyright permissions or are unsure of your material's copyright compatibility. Once you have responded and addressed all other outstanding technical requirements, you may resubmit your manuscript within Editorial Manager. Potential Copyright Issues: i) Figures 1, 3B, 4, and 5. Please confirm whether you drew the images / clip-art within the figure panels by hand. If you did not draw the images, please provide (a) a link to the source of the images or icons and their license / terms of use; or (b) written permission from the copyright holder to publish the images or icons under our CC BY 4.0 license. Alternatively, you may replace the images with open source alternatives. See these open source resources you may use to replace images / clip-art:- https://commons.wikimedia.org- https://openclipart.org/. ii) Figure 1 contains logos. We are not permitted to publish this under our CC BY 4.0 license,even with permission. We ask that you please remove or replace it. iii) You stated in the legend of Figure 2 that "The permanent URL for accessing the workflow in the PWB platform is: https://playbook-workflow-builder.cloud/report/d94b8b0a-81cc-708c-e200-e00ef3451da0." Please note that this licensed under CC BY-NC-SA . Unfortunately, the CC-BY-NC-SA license is not compatible with our CC BY 4.0 license as it has additional restrictions. Please amend the figure so that it is used from an openly available source. 4) Please amend your detailed Financial Disclosure statement. This is published with the article. It must therefore be completed in full sentences and contain the exact wording you wish to be published. 1) State the initials, alongside each funding source, of each author to receive each grant. For example: "This work was supported by the National Institutes of Health (####### to AM; ###### to CJ) and the National Science Foundation (###### to AM)." 2) State what role the funders took in the study. If the funders had no role in your study, please state: "The funders had no role in study design, data collection and analysis, decision to publish, or preparation of the manuscript.".If you did not receive any funding for this study, please simply state: u201cThe authors received no specific funding for this work.u201d 5)Thank you for stating that “There are no primary data in the paper; the project’s source code with instructions is available from: https://github.com/MaayanLab/Playbook-Workflow-Builder .” We notice that there is a CC BY-NC-SA license on your data. We would encourage you to consider using a license that is no more restrictive than CC BY, in line with PLOS’ recommendation on licensing (http://journals.plos.org/plosone/s/licenses-and-copyright)"http://journals.plos.org/plosone/s/licenses-and-copyright) .” **Reviewers' comments:**Reviewer's Responses to Questions

**Comments to the Authors:**

Reviewer #1: Hi, I am Stian Soiland-Reyes https://orcid.org/0000-0001-9842-9718 and have pledged the Open Peer Review Oath <https: 10.12688="" doi.org="" f1000research.5686.2="">:

* Principle 1: I will sign my name to my review

* Principle 2: I will review with integrity

* Principle 3: I will treat the review as a discourse with you; in particular, I will provide constructive criticism

* Principle 4: I will be an ambassador for the practice of open science

This review is licensed under a Creative Commons Attribution 4.0 International License http://creativecommons.org/licenses/by/4.0/

and is also available at the (for now) *secret* URL https://gist.github.com/stain/7751551d9dd64ea31a410a0fa0dc7348

## Recommendation</https:><https: 10.12688="" doi.org="" f1000research.5686.2=""></https:>

<https: 10.12688="" doi.org="" f1000research.5686.2="">

This article presents a workflow system for bioinformatics based on microservices. The authors have created a rich set of bindings for bionformatics tasks and a type-based system for their composition. A series of examples are described well for how the system can be used, but as I am not a bioinformatician by training I have not reviewed their scientific validity, and below rather I focus on workflow functionality of Playbook.

My main question is, given this would become entry 358 on

https://s.apache.org/existing-workflow-systems why we need another workflow system. The article hints at this, but does not make it clear. Galaxy is widely used and referenced by the article, along with perhaps more techie Snakemake and Nextflow; however the article shows little reflection on what makes Playbook different.

The creation of metanodes for adding tools to Playbook seems similar to adding to Galaxy's toolshed, which currently have over 10.000 tools https://toolshed.g2.bx.psu.edu/ The way of building-executing workflows step by step in Playbook is also clearly inspired by Galaxy, where a workflow can be retrospectively constructed from a data-driven history.

The authors claim workflow systems like Taverna were replaced by cloud-centric approaches. As former Taverna developer I would expand on this to mean a move from using public Web services to cloud/cluster based command line executions, which I will happily admit gave users greater scalability, stability and portability. Galaxy's tools for instance are primarily command line wrappers but some (particularly for data access) are REST API clients. The use of containers have also greatly improved the installation of such tools. But from this argument falls down as Playbook suggests using microservice architecture. On inspection of the source code I see that several metanodes are Node or Python-based, with dependencies installed into the main Playbook container, this is positive and could be made more clear in the manuscripts.

It is unclear from the paper who is hosting and maintaining the microservices that Playbook relies on, and this should be a concern for reproducibility and longer term availability. In a sense the author's proposed approach is moving back to the Web Service model of Taverna, but with a central knowledge graph of consise types rather than looser typed BioCatalogue (now subsumed by bio.tools which use EDAM types).

If the Playbook system can itself dynamically deploy these microservices (the article does not say), similar to Galaxy's tools spinning up containers, then the longevity of Playbook workflows could look brighter.

I would have liked to see more about how the Playbook type system was developed, and contrast it with earlier work of creating a single type-safe ecosystem of bioinformatics services, such as BioConductor, caGrid, BioMoby, SADI. Publications on Workflow Decay are particular relevant in this respect.

The authors should be commended for implementing export to standards BCO, CWL, RO-Crate. This use of FAIR Computational Workflow serialisations can significantly help against longer term workflow decay and improve Reusability of Playbook workflows. However, if the CWL steps simply call public web microservices, these exports are just as vulnerable to service changes/downtime, and this should be admitted as a potential risk. They would also not benefit from cloud scalability on CWL sode. It is not clear which of the metadata annotations e.g. license are carried on to these formats, here again there is great potential.

It is not clear from article if Playbook workflows are or can be published in WorkflowHub, this is certainly achievable with RO-Crate export, but may require an equivalent import on Playbook side, e.g. embedding its native JSON definition of the workflow template,

The Workflow Run Crate profile has been adopted recently by Galaxy and CWL engines, and could be considered for more detailed provenance records beyond BCO and Workflow Crate features used by WorkflowHub, see our recent paper https://doi.org/10.1371/journal.pone.0309210

Upon trying the Playbook, several very neat features seem to not be highlighted by the article:

- The user can document the workflow in a similar style to a Jupyter note book, e.g. adding references

- Interactive visualisations e.g. LINCS, Enrichr

- Example values

- View In Graph navigation

- LLM integration not described much

It was unclear to me if the Use Cases listed are meant to be reproducible, as I can't find distinction between a workflow template, a previous run, and my own rerun. Below are my observations:

- Use case 13: <https: 63d937db-8082-78f3-4961-57f814a0d0ef="" playbook-workflow-builder.cloud="" report=""> Input (example.h5ad?) is missing. Intermediate values are shown for the remaining steps, with visualisations. Final step Search PlyGen shows "Error: No matching protein product found."

- use case 1: <https: 6a7af61b-ef0a-7687-5d6e-02deeb253172="" playbook-workflow-builder.cloud="" report="">: All example inputs are provided. Clicking "Recompute" on various steps do not seem to do anything.

- Use case 11: <https: 24f9b280-82fa-4fb3-94e9-1ad80c7b9f3e="" playbook-workflow-builder.cloud="" report=""> Using the provided example "ACE2" works for the first steps, but fails in "Graph Plot". "MetGENE Metabolites with Gene Set" fails with `SyntaxError: Unexpected token.. DOCTYPE ... is not valid JSON". "MetGENE Studies Table" table is empty.

- Use case 27: Does not seem to be public listed?

I could verify BCO export and it imported well in <https: biocomputeobject.org="" builder=""> e.g. it shows individual steps. I exported the CWL and it included all annotations.

I was unable to find the RO-Crate export in the UI. This seems to be not yet merged into the main source code <https: 176="" github.com="" maayanlab="" playbook-workflow-builder="" pull=""> as the API endpoint <https: 86666a71-8d96-327e-2e7b-c2615b60cfca="" api="" playbook-workflow-builder.cloud="" ro-crate="" v1=""> does not work.

The manuscript should be modified to reflect that RO-Crate export is experimental branch. From a cursory look, the RO-Crate is quite sensible and includes links to the native JSON, CWL and BCO outputs; however it is almost compatible WorkflowHub <https: 1.0="" w3id.org="" workflow-ro-crate="" workflowhub=""> as the workflow files would be by URI references rather than embedded within a Zip file -- this output as a direct RO-Crate Metadata JSON from an API is what we now called Detached RO-Crate and will be part of RO-Crate 1.2 <https: 1.2-draft="" ro-crate="" specification="" structure.html="" www.researchobject.org="">

The source code of Playbook has been made available at <https: github.com="" maayanlab="" playbook-workflow-builder=""> but is not with an open source software license. Creative Commons License CC-BY-NC-SA 4.0 is NOT RECOMMENDED for software <https: creativecommons.org="" faq=""> and NC is not open source as it restricts commercial use.

Data availability statement should be modified to note that the software is not open source. Alternatively, a GNU license like GNU Affero General Public License <https: agpl-3.0.en.html="" licenses="" www.gnu.org=""> may be appropriate if the authors want to prevent commercial use without open source any modifications. You should check the licenses of your Python and NodeJS dependencies for compatibility with your chosen license.

I recommend enabling Zenodo archiving automatic from GitHub release <https: archiving-a-github-repository="" docs.github.com="" en="" referencing-and-citing-content="" repositories=""> -- this would give a code citation DOI persistent identifier authors can cite from the manuscript (in case GitHub goes).

## Summary

I have raised some other page-by-page suggestions below.

Overall I welcome this contribution, but would primarily ask the authors to point out contrasts to other systems for potential consumers. Table S.1 is very incomplete and is more of an illustration, the paper should rather reference review papers and guides like https://doi.org/10.1016/j.future.2017.01.012
https://doi.org/10.1016/j.patter.2021.100322
https://doi.org/10.1038/s41592-021-01254-9
https://doi.org/10.1007/978-1-4939-9074-0_24
https://doi.org/10.1371/journal.pcbi.1008622

I also have reservations with regards to reproducibility/rerunnability for a microservice-based environment, as I was unable to reproduce the listed use cases fully. A section should be added to discuss this.

The RO-Crate support should either be merged or described as experimental, so that the live website matches the manuscript. (I recommend merge and name it Detached RO-Crate).

Most of the citations are missing DOIs, ensure these are included in the print version.

Citation 15 for CWL specification should be replaced with journal paper <https: 10.1145="" 3486897="" doi.org="">

Citation 9-11 for Galaxy should include newest publication <https: 10.1093="" doi.org="" gkae410="" nar="">

## Detailed comments

p7. "although mostly underdeveloped, bioinformatics workflow engines... already exists"

This claim is unfounded; if anything, the most mature workflow engines are developed for bioinformatics.

Remove "Mostly underdeveloped".

p7. Clarify that Table S1 is not an extensive list but a sample.

S1 includes Taverna, but is missing X for "Graphical interface for workflow construction" and "Pre-defined modules". See <https: apache_taverna="" en.wikipedia.org="" wiki="">

S1 is using PubMed ID instead of DOI for citations, meaning that citations are missing for CWL <https: 10.1145="" 3486897="" doi.org="">, WDL <https: 10.7490="" doi.org="" f1000research.1114634.1="">, WorkflowHub <https: 10.5281="" doi.org="" zenodo.4605654="">, taverna <https: 10.1093="" doi.org="" gkt328="" nar="">

p7. "... Galaxy..Snakemake..Nextflow..... These newer platforms rely on ... The two leading standards for workflow languages.. CWL..WDL"

This seem to apply that Galaxy, Snakemake and Nexflow support CWL and WDL, however, only Galaxy has (experimental) support for CWL. Rephrase, e.g. "In addition, two leading standards.."

p7. "Other examples of community standards ... BioCompute... and WorkflowHub"

WorkflowHub is not a community standard, but a registry. The community standard used by WorkflowHub is (as the next sentence explains), RO-Crate.

In particular, WorkflowHub uses a profile of RO-Crate that is called Workflow RO-Crate https://w3id.org/workflowhub/workflow-ro-crate/1.0

p7. Federated Knowledge Graphs: Considerations of former work could mention BioConductor <https: 10.1186="" doi.org="" gb-2004-5-10-r80="">, BioMoby <https: 10.1093="" 3.4.331="" bib="" doi.org=""> <https: 10.1016="" doi.org="" j.jbi.2008.02.005="">, SADI <https: 10.1007="" 978-3-642-16558-0_26="" doi.org=""> <https: 10.1109="" apscc.2009.5394148="" doi.org="">, caBIG <https: 10.1197="" doi.org="" jamia.m2522=""> <https: bringing-cabig-services-together-using-taverna="" en="" publications="" research.manchester.ac.uk=""> -- these are all at least a decade old.

p8. add citations or urls for CF and CFDE.

p9. persistent urls are not exposed in the user interface as such. A persistent url should use a PID provider, e.g. DataCite/DOI, https://w3id.org/ or https://purl.archive.org/ -- urls shown in the interface are called "temporary links", indicating they are _not_ persistent?

p13 - p15. Please add the persistent urls for each use case (see above)

p18. "operated-on at runtime in a type-safe manner through runtime-based type checkin" -- avoid repetition of "type" and "runtime"</https:></https:></https:></https:></https:></https:></https:></https:></https:></https:></https:></https:></https:></https:></https:></https:></https:></https:></https:></https:></https:></https:></https:></https:></https:></https:></https:>

Reviewer #2: This paper introduces a meaningful contribution to bioinformatics by addressing current challenges in workflow integration, usability, and reproducibility. PWB leverages a comprehensive ecosystem of APIs and a novel KRG framework, offering a scalable solution to handle diverse bioinformatics data types. The inclusion of 561 metanodes provides flexibility, and the capacity to visualize results and reuse workflows enhances the utility of the platform for both novice and experienced users.

This paper includes strengths in terms of the innovation in workflow construction, standards compliance, and real-world application. The platform’s use of a chatbot powered by a large language model (LLM) for workflow guidance stands out as an innovative approach to reduce technical barriers, making the platform accessible. By aligning with CWL and BCO standards, PWB supports workflow portability and interoperability, which are crucial for collaborative and reproducible research. The detailed use cases demonstrate PWB's potential in handling real-world bioinformatics challenges, showcasing its adaptability across different biomedical research domains.

There are some concerns.

1. Performance evaluation using empirical benchmarks could make this paper persuasive. Can you compare PWB's efficiency and accuracy with other workflow tools, such as Galaxy or Snakemake?

2. While the platform's scalability is theoretically addressed, more details on its performance under high data loads or with complex workflows would be beneficial.

3. User studies or feedback on the interface and workflow creation experience would help substantiate the claim that PWB is user-friendly for non-experts.

**Have the authors made all data and (if applicable) computational code underlying the findings in their manuscript fully available?**

Reviewer #1: Yes

Reviewer #2: Yes

PLOS authors have the option to publish the peer review history of their article (what does this mean? ). If published, this will include your full peer review and any attached files.

**Do you want your identity to be public for this peer review?** For information about this choice, including consent withdrawal, please see our Privacy Policy .

Reviewer #1: **Yes: ** Stian Soiland-Reyes

Reviewer #2: No

**Figure resubmission:**While revising your submission, please upload your figure files to the Preflight Analysis and Conversion Engine (PACE) digital diagnostic tool, https://pacev2.apexcovantage.com/. PACE helps ensure that figures meet PLOS requirements. To use PACE, you must first register as a user. Registration is free. Then, login and navigate to the UPLOAD tab, where you will find detailed instructions on how to use the tool. If you encounter any issues or have any questions when using PACE, please email PLOS at figures@plos.org. Please note that Supporting Information files do not need this step. If there are other versions of figure files still present in your submission file inventory at resubmission, please replace them with the PACE-processed versions.
---

## [Decision Letter · Decision Letter 1]

24 Feb 2025

Dear Dr. Ma'ayan,

We are pleased to inform you that your manuscript 'Playbook Workflow Builder: Interactive Construction of Bioinformatics Workflows' has been provisionally accepted for publication in PLOS Computational Biology.

Best regards,

Pedro Mendes, PhD

Section Editor

PLOS Computational Biology

Pedro Mendes

Section Editor

PLOS Computational Biology

Reviewer's Responses to Questions

**Comments to the Authors:**

Reviewer #1: The authors have made significant changes to the manuscript reflecting previous reviewer comments, I welcome these improvements with a new reproducibility section, and I am pleased with the extra effort like archiving code in Zenodo and WorkflowHub.

The deposited Workflow RO-Crate in WorkflowHub https://doi.org/10.48546/WORKFLOWHUB.WORKFLOW.1237.2 has a 0 byte workflow file "7a420ac4-96c0-e7e3-36cb-0d56babbafad", this should be fixed (replace with "export" file?).

The exported CWL file in WorkflowHub references other CWL files like "EnrichrTermSearch[Drug].cwl" which are missing from WorkflowHub deposit, these should be added as otherwise the workflow will neither run nor parse. I would recommend building on the "CWL bundle" as you export it from the platform, this would include all the details CWL-wise and would combine well with the JSON to make an RO-Crate.

The RO-Crate JSON exported from the platform differs significantly from the one uploaded to WorkflowHub. First of all it is in "detached" mode which is OK but should be called "RO-Crate (Detached)". A corresponding "RO-Crate (Attached)" would be within a ZIP file with the other files in addition to the JSON, as in the WorkflowHub entry or the CWL bundle.

While the WorkflowHub entry is valid, the export is not, as it is not in Flattened JSON-LD form as detailed in https://www.researchobject.org/ro-crate/specification/1.1/structure.html#ro-crate-metadata-file-ro-crate-metadatajson and https://www.researchobject.org/ro-crate/specification/1.1/appendix/jsonld.html -- I found that simply using the JSON-LD flattening algorithm was sufficient to fix this.

As these are my suggested fixes to the platform rather than the article, I will recommend to Accept the manuscript.

Reviewer #2: All my comments have been thoroughly addressed in this revision. There are a couple of typos in the manuscript (e.g. one in the third paragraph of the Introduction section and another in the second paragraph of section “Use Case 1: Explaining Drug-Drug Interactions”). If these typos are corrected, I would be happy to see this paper published.

**Have the authors made all data and (if applicable) computational code underlying the findings in their manuscript fully available?**

Reviewer #1: Yes

Reviewer #2: Yes

PLOS authors have the option to publish the peer review history of their article (what does this mean? ). If published, this will include your full peer review and any attached files.

**Do you want your identity to be public for this peer review?** For information about this choice, including consent withdrawal, please see our Privacy Policy .

Reviewer #1: **Yes: ** Stian Soiland-Reyes

Reviewer #2: No

---

## [Editor Report · Acceptance letter]

PCOMPBIOL-D-24-01433R1

Playbook Workflow Builder: Interactive Construction of Bioinformatics Workflows

Dear Dr Ma'ayan,

I am pleased to inform you that your manuscript has been formally accepted for publication in PLOS Computational Biology. Your manuscript is now with our production department and you will be notified of the publication date in due course.

With kind regards,

Anita Estes
